# *Pseudomonas* Bacteremia in a Tertiary Hospital and Factors Associated with Mortality

**DOI:** 10.3390/antibiotics12040670

**Published:** 2023-03-29

**Authors:** Petros Ioannou, Konstantinos Alexakis, Sofia Maraki, Diamantis P. Kofteridis

**Affiliations:** 1School of Medicine, University of Crete, 71003 Heraklion, Greece; 2Internal Medicine Department, University Hospital of Heraklion, 71110 Heraklion, Greece; 3Department of Clinical Microbiology, University Hospital of Heraklion, 71110 Heraklion, Greece

**Keywords:** *Pseudomonas*, bacteremia, bloodstream infection, antimicrobial resistance, mortality

## Abstract

*Pseudomonas aeruginosa* is the third most commonly identified cause among gram-negative microorganisms causing bloodstream infection (BSI) and carries a very high mortality, higher than that by other gram-negative pathogens. The aim of the present study was to assess the epidemiological and microbiological characteristics of patients with BSI by *Pseudomonas* spp. in a tertiary hospital, characterize the resistance rates of different *Pseudomonas* strains to the most clinically relevant anti-microbials, estimate the mortality rate, and identify factors independently associated with mortality. In total, 540 cultures from 419 patients sent to the microbiology department of the hospital during the 8-year period of the study were positive. Patients had a median age of 66 years, and 262 (62.5%) were male. The blood culture was drawn in the ICU in 201 of the patients (48%). The infection was hospital-acquired in 329 patients (78.5%) and the median hospital day when the blood culture was drawn was 15, with a range of 0 to 267 days. Median duration of stay in the hospital was 36 days, hospital mortality was 44.2% (185 patients), and 30-day mortality was 29.6% (124 patients). The most commonly isolated *Pseudomonas* species were *P. aeruginosa* followed by *P. putida* and *P. oryzihabitans*. There was a statistically significant reduction of *P. aeruginosa* isolation relative to non-*aeruginosa Pseudomonas* species in the post-COVID-19 era. Antimicrobial resistance of *P. aeruginosa* in clinically relevant antimicrobials with anti-pseudomonal activity was similar before and after the onset of the COVID-19 pandemic with the exception of gentamicin and tobramycin, with *P. aeruginosa* being more susceptible to these two antimicrobials in the post-COVID-19 era. Rates of multi-drug resistant (MDR), extensively-drug resistant (XDR), and difficult-to-treat (DTR) *P. aeruginosa* isolation were lower after the onset of the COVID-19 pandemic, even though a carbapenem-focused antimicrobial stewardship program had been implemented in the meantime. Increased age, ICU-acquisition of BSI, and more days in the hospital when positive blood culture was drawn were positively associated with 30-day mortality of patients with *Pseudomonas* BSI. The fact that rates of MDR, XDR, and DTR *P. aeruginosa* isolation were lower late in the study period, with a carbapenem-focused antimicrobial stewardship intervention being implemented in the meantime, further increases the understanding that implementation of antimicrobial stewardship interventions may halt the increase in antimicrobial resistance noted previously.

## 1. Introduction

*Pseudomonas aeruginosa* is the third most commonly identified cause among gram-negative microorganisms causing bloodstream infection (BSI) and carries a very high mortality, higher than that by other gram-negative pathogens and also higher than that of *Staphylococcus aureus,* and is estimated to be at 30% after 30 days from the BSI episode [1,2,3,4]. BSI by *Pseudomonas* spp. is a universal problem and is more commonly hospital-acquired [5,6,7,8,9]. In specific settings, such as in intensive care units (ICU), the rate of *Pseudomonas* spp. isolation as pathogens causing BSIs is even higher. For example, in a study about ICU-acquired BSI, *Pseudomonas aeruginosa* was the second most common cause of BSI among gram-negative pathogens [10]. In another study regarding patients with liver transplantation, 35% of BSIs were due to *P. aeruginosa* and had an attributed mortality of 30% [11]. Commonly described risk factors for bacteremia by *P. aeruginosa* include recent antimicrobial use (most commonly involving the last three months), recent hospitalization, severe burns, pancreatobiliary disease, neutropenia or other immunodeficiency (such as hematologic malignancy or transplantation), and presence of an indwelling central venous or urinary catheter [6,8,12,13,14,15,16].

The coronavirus-disease 2019 (COVID-19) pandemic has caused pressure in healthcare systems, has led to increased rates of hospitalization, increased antimicrobial consumption, and has also led to a higher prevalence of multi-drug-resistant (MDR) pathogens [17,18]. This could be associated with increased rates of hospitalization, and at least to some extent, unnecessary antimicrobial use, healthcare personnel exhaustion, and possibly, a relative reduction in the implementation of antimicrobial stewardship practices [19]. Thus, due to the high mortality associated with *Pseudomonas* BSIs and the changing landscape in antimicrobial resistance and incidence of these infections in the post-COVID-19 era, studies evaluating the characteristics of these infections are warranted.

The aim of the present study was to assess the epidemiological and microbiological characteristics of patients with BSI by *Pseudomonas* spp. in a tertiary hospital, characterize the resistance rates of different *Pseudomonas* strains to the most clinically relevant antimicrobials, estimate the mortality rate, and identify factors independently associated with mortality.

## 2. Materials and Methods

### 2.1. Study Type and Ethics Approval

This is a retrospective single-center study including data regarding all cases of BSIs due to *Pseudomonas* spp. from patients hospitalized from 2015 to 2022 in the University Hospital of Heraklion, Heraklion, Greece, a tertiary hospital with 771 beds. All data were retrieved, retrospectively, from the database of the Department of Microbiology and were then evaluated. Data from blood cultures were included if the culture was positive for growth of *Pseudomonas* spp. Data collected and evaluated included type of sample that yielded a positive culture, date the culture was collected, microorganism identified, and antimicrobial resistance of the isolated microorganisms. Further information was extracted from the electronic health records regarding patients’ age, gender, duration of hospitalization, day of hospitalization the positive culture was taken, ward where the blood was drawn, outcome of hospitalization, and outcome at 30-days after the bacteremia. MDR and extensively-drug-resistant (XDR) pathogens were defined as previously published [20]. Difficult-to-treat (DTR) *P. aeruginosa* was defined as the strain resistant to all beta-lactams tested and fluoroquinolones as previously described [21]. Infection was considered hospital-acquired if the blood culture was drawn at least 48 h after admission. Post-COVID-19 era was defined as the era from 2020 until the end of the study. Conduction of the study was approved by the Institutional Review Board of the University Hospital of Heraklion.

### 2.2. Sample Collection, Transport, and Processing

Blood cultures were obtained from patients to investigate the possibility of bacteremia. Blood was collected in blood culture bottles that were promptly transported to the microbiology laboratory for further processing. Bottles were loaded and incubated on the BacT/Alert Virtuo system (BioMérieux, Marcy L’ Etoile, France) for five days unless growth was detected earlier. When a culture bottle signaled positive, gram stain and subcultures were then immediately performed. For the isolation of bacterial pathogens, specimens were inoculated onto Columbia blood, chocolate, MaC Conkey, and Schaedler blood agar (all products of bioMérieux SA, Marcy L’Étoile, France) and incubated at 36 °C. Identification of bacterial species was performed by standard biochemical assays and the Vitek 2 automated system and confirmed by the matrix-assisted laser desorption time of flight mass spectrometry (MALDI-TOF MS) (version 3.2) (both products of bioMérieux SA, Marcy L’Étoile, France). The Vitek 2 automated system was also used for antibiotic susceptibility testing and results were interpreted according to the 2021 Clinical and Laboratory Standards Institute (CLSI) criteria [22]. As quality control strains, *Escherichia coli* ATCC 25922, *Pseudomonas aeruginosa* ATCC 27853, *S. aureus* ATCC 25923, and *E. faecalis* ATCC 29212 were used.

### 2.3. Statistics

Descriptive statistics were performed with GraphPad Prism 6.0 (GraphPad Software, Inc., San Diego, CA, USA). Qualitative data were presented as counts and percentages. Categorical data were analyzed with Fisher’s exact test. Continuous variables were compared using Student’s t-test for normally distributed variables and the Mann–Whitney U-test for non-normally distributed variables. All tests were two-tailed and *p* ≤ 0.05 was considered to be significant. Data are presented as numbers (%) for categorical variables and medians (interquartile range (IQR)) or means (±standard deviation (SD)) for continuous variables. A linear-regression analysis model was developed to evaluate the effect of several parameters (age, gender, *P. aeruginosa* BSI, antimicrobial resistance to specific antimicrobials, ward where positive blood culture was taken, day of hospitalization when positive blood culture was taken) with 30-day mortality. All the parameters mentioned above were calculated with GraphPad Prism 6.0 (GraphPad Software, Inc., San Diego, CA, USA). A multivariate logistic-regression analysis model was developed to evaluate the association of factors identified in the univariate analysis with a *p* ≤ 0.05 with mortality. Multivariate analysis was performed using the SPSS version 23.0 (IBM Corp., Armonk, NY, USA).

## 3. Results

### 3.1. Patients’ Characteristics

In total, 540 cultures from 419 patients that had been sent to the microbiology department of the hospital during the 8-year period of the study were positive. Patients had a median age of 66 years and 262 (62.5%) were male. The blood culture was drawn in the ICU in 201 (48%), in a medical ward in 142 (33.9%), and in a surgical ward in 76 (18.1%). The infection was hospital-acquired in 329 patients (78.5%) and the median hospital day the blood culture was drawn was 15 with a range of 0 to 267 days. Median duration of stay in the hospital was 36 days, and hospital mortality was 44.2% (185 patients), while mortality at 30-days after occurrence of bacteremia was 29.6% (124 patients).

Patients who died up to 30-days after the occurrence of bacteremia had a higher age, were more likely to have a hospital-acquired BSI, and more specifically, were more likely to have acquired the BSI later in their hospitalization, were less likely to be hospitalized in a surgical ward, more likely to be hospitalized in the ICU, and were more likely to have a longer duration of hospitalization. Table 1 shows the characteristics of patients with *Pseudomonas* spp. and shows a comparison among patients who survived and those who died up to 30-days after the occurrence of bacteremia.

### 3.2. Microbiology of Pseudomonas BSIs

The most commonly isolated *Pseudomonas* species were *P. aeruginosa* followed by *P. putida* and *P. oryzihabitans*. Appendix A shows the microbiology of the isolated species in total as well as the year of the study, while Table 2 shows the microbiology of *Pseudomonas* species isolated in total as well as whether the episode of BSI occurred in the pre-COVID-19 era or the post-COVID-19 era. Occurrence of BSIs by *Pseudomonas* spp. was relatively stable during the years of the study. There was a statistically significant reduction of *P. aeruginosa* isolation compared to non-aeruginosa *Pseudomonas* species in the post-COVID-19 era.

A comparison of the microbiology of *Pseudomonas* BSIs among patients who had community-acquired BSI compared to that of patients with hospital-acquired BSI shows a statistically significant higher rate of *P. putida* isolation relative to other *Pseudomonas* species in patients who had a hospital-acquired BSI. Appendix A shows the microbiology of *Pseudomonas* strains isolated from patients’ blood cultures as well as whether the BSI was acquired in the hospital or in the community.

Appendix A shows the microbiology of *Pseudomonas* strains isolated from patients’ blood cultures as well as whether the patient was hospitalized in a ward or in the intensive care unit at the time the blood culture was drawn. A comparison among these different groups of patients did not reveal any statistically significant difference.

A comparison of the microbiology of *Pseudomonas* BSIs among patients who survived with those who died shows a statistically significant higher rate of *P. aeruginosa* isolation relative to non-aeruginosa *Pseudomonas* species in patients who died. Table 3 shows the microbiology of *Pseudomonas* strains isolated from patients’ blood cultures compared to patients’ outcome at 30-days after the occurrence of bacteremia.

### 3.3. Antimicrobial Resistance of P. aeruginosa Isolated from BSI Episodes

Antimicrobial resistance of *P. aeruginosa* in clinically relevant antimicrobials with anti-pseudomonal activity was similar before and after the onset of the COVID-19 pandemic with the exception of gentamicin and tobramycin, with *P. aeruginosa* being more susceptible to these two antimicrobials in the post-COVID-19 era in a statistically significant manner. Table 4 shows the antimicrobial resistance to *P. aeruginosa* in total as well as whether isolation was before or after the onset of the COVID-19 pandemic. Interestingly, in the post-COVID-19 era, isolation of MDR, XDR, and DTR strains was less frequent.

A comparison of antimicrobial resistance of *P. aeruginosa* strains among patients who had community-acquired BSI with those who had hospital-acquired BSI showed a higher resistance of *P. aeruginosa* to aztreonam, meropenem, and ticarcillin in strains isolated from hospital-acquired BSIs. Appendix A shows the antimicrobial resistance of *P. aeruginosa* strains isolated from patients’ blood cultures as well as whether the BSI was community- or hospital-acquired.

A comparison of antimicrobial resistance of *P. aeruginosa* strains among patients hospitalized in the wards compared to those who were hospitalized in the ICU at the time the blood culture was drawn did not reveal any statistically significant differences. Appendix A shows the antimicrobial resistance of *P. aeruginosa* strains isolated from patients’ blood cultures compared to where the patient was hospitalized at the time the blood culture was drawn.

A comparison of antimicrobial resistance of *P. aeruginosa* strains among patients who survived with those who died showed that patients who died had more resistant strains causing BSI in a statistically significant way for all antimicrobials except for colistin. Table 5 shows the antimicrobial resistance of *P. aeruginosa* strains isolated from patients’ blood cultures compared to patients’ outcome at 30-days after the occurrence of bacteremia.

Appendix A shows the antimicrobial resistance of the different *Pseudomonas* species isolated during the study period. *P. aeruginosa* was the most resistant species among the clinically relevant anti-pseudomonal antimicrobials followed by *P. putida* and *P. stutzeri*.

### 3.4. Regression Analysis of 30-Day Mortality among Patients with BSI by Pseudomonas spp.

To identify factors associated with mortality at 30-days after the occurrence of bacteremia, we performed regression analysis as follows. First, we performed univariate linear-regression analysis to evaluate the effect of several parameters such as age, gender, *P. aeruginosa* BSI, pattern of antimicrobial resistance, ward where positive blood culture was taken, and day of hospitalization when positive blood culture was taken with mortality at 30-days after the occurrence of bacteremia. Age, ICU-acquisition, more days in the hospital when the positive blood culture was taken, *P. aeruginosa* BSI, MDR, XDR, and DTR phenotype were all found to be positively associated with mortality at 30-days after the occurrence of bacteremia. However, a multivariate logistic-regression model identified only increased age (*p* < 0.001, odds ratio 1.032), ICU-acquisition (*p* = 0.01, odds ratio 2.213), and more days in the hospital when positive blood culture was taken (*p* = 0.005, odds ratio 1.01) to be independently positively associated with mortality at 30-days after the occurrence of bacteremia. Table 6 shows the results of the regression analysis of 30-day mortality among patients with *Pseudomonas* spp. BSI.

## 4. Discussion

The present study investigated the microbiology and antimicrobial resistance of *Pseudomonas* spp. BSI in a tertiary hospital in Crete, Greece. The most common species was *P. aeruginosa,* which was also the most resistant to antimicrobials. Isolation of *Pseudomonas* spp. was stable throughout the study period. Mortality at 30-days after occurrence of BSI was high and patients who died had a higher age, were more likely to have hospital-acquired BSI, and more specifically, were more likely to have acquired the BSI later in their hospitalization, were less likely to be hospitalized in a surgical ward and more likely to be hospitalized in the ICU, were more likely to have a longer duration of hospitalization, and were also more likely to have a BSI by *P. aeruginosa* as well as increased antimicrobial resistance. Increased age, ICU-acquisition of BSI, and more days in the hospital when positive blood culture was drawn were independently positively associated with 30-day mortality.

BSIs by *Pseudomonas* spp. and more specifically, by *P. aeruginosa*, are commonly encountered in clinical practice, especially in patients with risk factors for acquisition of this pathogens, such as advanced age, neutropenia or other immunodeficiency (e.g., malignancy or transplantation), indwelling central venous or urinary catheter, severe burns, pancreatobiliary disease, previous receipt of antimicrobials, and previous or current hospitalization [5,6,7,8,9,12,16,23]. The clinical presentation includes the typical features noted for all patients with gram-negative sepsis, such as fever, tachycardia and tachypnea, hypotension, respiratory failure, and disorientation, even though these findings may also occur in cases of sepsis by other organisms as well, and they are not specific exclusively for gram-negative sepsis. The most common sources of *P. aeruginosa* sepsis and bacteremia include the respiratory tract, infected intravascular catheters, the gastrointestinal and hepatobiliary tract, the urinary tract, and the skin and soft tissues. However, in up to 40% of cases, the source of bacteremia is not evident [7,24,25].

In the present study, the most common species, as anticipated, was *P. aeruginosa*, which is the most clinically relevant species causing infection, and more specifically BSIs, in humans. Indeed, most studies on BSIs refer exclusively to *P. aeruginosa* so that it may be hard to compare data on non-*aeruginosa* species to others in the literature. A study referring to non-*aeruginosa Pseudomonas* species in respiratory samples of patients with cystic fibrosis identified *P. fluorescens*, *P. putida*, and *P. stutzeri* as the most common species with 33, 18, and 6 isolated strains, respectively, followed by *P. alcaligenes*, *P. fragi*, *P. mendocina*, *P. nitroreducens*, *P. oleovorans*, *P, oryzihabitans*, and *P, veronii*, each with one isolated stain [26]. In the same study, the clinical importance of the non-*aeruginosa Pseudomonas* species is discussed, suggesting that some of them may be of low pathogenicity. In the present study, the most commonly isolated species beyond *P. aeruginosa* were *P. putida*, *P. oryzihabitans*, and *P. stutzeri*. An older study evaluating non-*aeruginosa Pseudomonas* species as causes of BSI identified a small number of such species and discusses the clinical relevance their isolation may have, since most patients with such species did not receive antimicrobial treatment, with no mortality noted [27]. In the present study, even though there were no data regarding antimicrobial treatment of the patients, a statistically significant association between isolation of *P. aeruginosa* and 30-day mortality was noted, even though this was not identified as an independent factor associated with 30-day mortality in a multivariate logistic-regression analysis model.

Even though the number of *Pseudomonas* isolates identified in blood cultures remained stable through the years, there was a statistically significant reduction of *P. aeruginosa* strains in the post-COVID-19 era. Infection by non-*aeruginosa Pseudomonas* species is relatively uncommon and it mostly occurs in patients with underlying immunocompromise, or in association with an infected medical device [28,29,30].

Antimicrobial resistance among gram-negative microorganisms is a problem of increasing magnitude in medicine, associated with increased morbidity, mortality, and health costs [31,32]. Currently, for many pathogens, available antimicrobial treatments are limited, and choices include revived antibiotics or combinations of antimicrobials [33,34,35,36,37]. *P. aeruginosa* has many different mechanisms of antimicrobial resistance that may include production of antibiotic-inactivating enzymes, efflux systems, and modification of the permeability of the outer membrane [38]. In the present study, regarding antimicrobial resistance, *P. aeruginosa* has the highest rates of resistance to clinically relevant antimicrobials with anti-pseudomonal activity, such as piperacillin, ciprofloxacin, and cefepime. However, antimicrobial resistance to old antibiotics such as colistin is still low, meaning that even in cases of BSI by *P. aeruginosa* with multiple resistance mechanisms, there are still available options in terms of treatment. Nowadays, newer antimicrobials and antimicrobial combinations, such as the combination of ceftolozane with tazobactam, the combination of ceftazidime with avibactam, cefiderocol, the combination of imipenem with cilastatin and relebactam, or the combination of aztreonam with avibactam have emerged as important therapeutic options that should be preferred for the treatment of BSI by sensitive microorganisms such as *Pseudomonas* species [21,39,40,41]. In the present study, these newer antimicrobials were not tested; thus, no result can be drawn regarding their susceptibility in the isolated strains. Interestingly, other species such as *P. putida* also have high rates of antimicrobial resistance to the same antimicrobials. This is in line with studies showing that, even though non-*aeruginosa Pseudomonas* species may have been previously considered to have low pathogenicity, some of them, such as *P. putida,* are causing significant mortality in cases of BSI, and also have important antimicrobial resistance [30,42]. For example, *P. putida* has been identified as a nosocomial cause of infection with multi-drug-resistance and production of VIM-2 metallo-beta-lactamase, probably by independent horizontal transfer of resistance genes [43,44].

In the present study, antimicrobial resistance of *P. aeruginosa* seemed to have remained stable throughout the study period when studying each antimicrobial individually, with the exception of gentamicin and tobramycin, as, in the post-COVID-19 era, *P. aeruginosa* had less resistance against these two aminoglycosides. Interestingly, rates of MDR, XDR, and DTR were lower in patients with BSI by *P. aeruginosa* in the post-COVID-19 era. However, it is of note that, during the study period, a carbapenem-focused antimicrobial stewardship program was implemented and led to a reduction of use of carbapenems through unsolicited consultations of infectious disease physicians; this may have affected the time-course of development of antimicrobial resistance, by halting a possibly increasing trend [45,46]. This is of utmost importance since it increases the common understanding that implementation of antimicrobial stewardship interventions may lead to reduction of antimicrobial use, reduction of hospital costs, and reduction of antimicrobial resistance without an increase in mortality [47]. In some other studies, antimicrobial resistance of *P. aeruginosa* strains also remained stable during the COVID-19 pandemic [48,49,50]. More specifically, in some studies, an increase in infections by carbapenem-susceptible *P. aeruginosa* was noted, which could theoretically be associated with a higher consumption of third generation cephalosporins during the first wave of the COVID-19 pandemic [49]. In other studies, however, trends for antimicrobial resistance of *P. aeruginosa* was rising during the COVID-19 pandemic [51,52].

Interestingly, for all antimicrobials tested, with the single exception of colistin, there was a statistically significant association between antimicrobial resistance and 30-day mortality, while MDR, XDR, and DTR *P. aeruginosa* was more frequently isolated among patients with BSI who died within 30-days from the positive blood culture. However, this could have been a result of the higher likelihood of ICU-acquisition of *Pseudomonas* BSI among patients who died, as there was a non-statistically significant trend for higher antimicrobial resistance of *P. aeruginosa* strains isolated in the ICU. To that end, a multivariate logistic-regression analysis identified ICU-acquisition of *Pseudomonas* BSI to be independently associated with 30-day mortality, while antimicrobial resistance was not. Notably, the fact that diagnosis of the BSI in the ICU is associated with higher mortality is probably an epiphenomenon of the more severe clinical condition of the patients diagnosed in this setting. Indeed, patients admitted to the ICU have higher severity of illness and higher mortality when diagnosed with BSI [53]. However, since the present study is based mostly on microbiological data, a direct association of BSI mortality with the severity of the underlying illness addressed with clinical severity scores such as SOFA or APACHE could not have been performed.

The multivariate logistic-regression analysis identified also increased age and increased length of hospitalization before drawing the positive blood culture for *Pseudomonas* spp. to be independently associated with 30-day mortality. Increased age is well known to be associated with increased mortality in studies of hospitalized patients in general; thus, this finding was not a surprise [54]. On the other hand, length of hospitalization before the positive blood culture could also be an epiphenomenon, since length of hospitalization may also be a factor that could be theoretically associated with the clinical condition of the patient, as patients with more complex medical or surgical problems may have a need for more prolonged hospitalization and may have higher mortality, especially in the case of BSI by drug-resistant gram-negative pathogens [55]. However, the present study did not focus on the clinical characteristics of the patients; thus, this presumed association of length of hospitalization before drawing a blood culture in patients with BSI by *Pseudomonas* spp. and the severity of their clinical condition has not been confirmed in the present study. In other studies, though, prolonged hospitalization in general has also been associated with mortality [54].

This study has some limitations that should be acknowledged. First, it is a single-center study; thus, the results should be read cautiously as they represent the microbiology and antimicrobial resistance patterns of a specific area. Furthermore, as this study mostly included microbiological data, there are no data regarding patients’ clinical characteristics or treatment. For example, the criterion of hospital-acquired bacteremia was whether the blood culture was drawn after 48 h from admission, which may not necessarily coincide with the most appropriate definition of development of symptoms of infection after 48 h from the admission. Moreover, since clinical data were not collected, whether BSI episodes were primary or secondary was not addressed in the present study. On the other hand, clinical and laboratory data allowing estimation of the severity of the underlying disease through calculation of SOFA or APACHE scores was not possible in the present study. Moreover, the antimicrobials tested are not all that are available; thus, newer antimicrobial combinations, such as that of ceftazidime/avibactam or ceftolozane/tazobactam have not been included in the analysis. Finally, some of the *Pseudomonas* species were very few; thus, the data regarding antimicrobial resistance may not be reliable enough to draw safe conclusions.

## 5. Conclusions

BSIs by *Pseudomonas* spp. carry significant mortality. *P. aeruginosa* is the most commonly identified species and has significant antimicrobial resistance; however, other species such as *P. putida* may also resemble significant resistance. Increased age, ICU-acquisition of BSI, and more days in the hospital when positive blood culture was drawn were independently positively associated with 30-day mortality. The fact that rates of MDR, XDR, and DTR *P. aeruginosa* isolation were lower late during the study period, with a carbapenem-focused antimicrobial stewardship intervention being implemented in the meantime, further increases the understanding that implementation of antimicrobial stewardship interventions may halt the increase in antimicrobial resistance noted previously.

## Figures and Tables

**Table 1 antibiotics-12-00670-t001:** Characteristics of patients with *Pseudomonas* spp. bacteremia in all patients and in regard to mortality at 30-days.

	2015–2022 (*n* = 419)	Survived (*n* = 295)	Died (*n* = 124)	*p*
Age, years, median (IQR)	66 (48–76.3)	62 (39–75)	72 (59–80)	<0.0001
Male gender, *n* (%)	262 (62.5)	184 (62.4)	78 (62.9)	1.0000
Site where culture was collected				
Medical ward, *n* (%)	142 (33.9)	95 (32.2)	47 (37.9)	0.2610
Surgical ward, *n* (%)	76 (18.1)	71 (24.1)	5 (4.0)	<0.0001
ICU, *n* (%)	201 (48.0)	129 (43.7)	72 (58.1)	0.0076
Hospital-acquired, *n* (%)	329 (78.5)	61 (20.7)	124 (100)	<0.0001
Hospital day when the positive culture was drawn, median (IQR)	15 (4–32)	14 (3–27)	20 (5–43.8)	0.0089
Duration of hospital stay, days, median (IQR)	36 (19–71)	42.5 (20–76.3)	30.5 (15.3–51)	0.0008
Hospital mortality, *n* (%)	185 (44.2)			
30-day mortality, *n* (%)	124 (29.6)			

ICU: intensive care unit; IQR: interquartile range.

**Table 2 antibiotics-12-00670-t002:** Microbiology of *Pseudomonas* strains isolated from patients’ blood cultures.

Pathogen	2015–2022 (%)	Pre-COVID-19 (%)	Post-COVID-19 (%)	*p*
*P. aeruginosa*	357 (85.2)	232 (88.6)	125 (79.6)	0.0155
*P. alcaligenes*	2 (0.5)	0 (0.0)	2 (1.3)	0.1398
*P. fluorescens*	4 (1.0)	2 (0.8)	2 (1.3)	0.6325
*P. luteola*	1 (0.2)	0 (0.0)	1 (0.6)	0.3747
*P. mendocina*	2 (0.5)	1 (0.4)	1 (0.6)	1.0000
*P. oleovorans*	2 (0.5)	0 (0.0)	2 (1.3)	0.1398
*P. oryzihabitans*	17 (4.1)	11 (4.2)	6 (3.8)	1.0000
*P. putida*	26 (6.2)	12 (4.6)	14 (8.9)	0.0938
*P. stutzeri*	8 (1.9)	4 (1.5)	4 (2.6)	0.4799
All *Pseudomonas* strains	419 (100)	262 (62.5)	157 (37.5)	NA

COVID-19: coronavirus-disease 2019; NA: not applicable.

**Table 3 antibiotics-12-00670-t003:** Microbiology of *Pseudomonas* strains isolated from patients’ blood cultures, and patients’ outcome at 30 days after the occurrence of bacteremia.

Pathogen	Survived (%)	Died (%)	*p*
*P. aeruginosa*	243 (82.4)	114 (91.9)	0.0106
*P. alcaligenes*	2 (0.7)	0 (0.0)	1.0000
*P. fluorescens*	4 (1.4)	0 (0.0)	0.3239
*P. luteola*	1 (0.4)	0 (0.0)	1.0000
*P. mendocina*	1 (0.4)	1 (0.8)	0.5048
*P. oleovorans*	2 (0.7)	0 (0.0)	1.0000
*P. oryzihabitans*	13 (4.4)	4 (3.2)	0.7873
*P. putida*	22 (7.5)	4 (3.2)	0.1222
*P. stutzeri*	7 (2.4)	1 (0.8)	0.44456
All *Pseudomonas* strains	243 (70.4)	124 (29.6)	NA

NA: not applicable.

**Table 4 antibiotics-12-00670-t004:** Antimicrobial resistance of *Pseudomonas aeruginosa* strains isolated from patients’ blood cultures.

Antibacterial	2015–2022 (%)	Pre-COVID-19 (%)	Post-COVID-19 (%)	*p*
Amikacin	61 (17.1)	45 (19.4)	16 (12.6)	0.1081
Aztreonam	90 (25.3)	62 (26.8)	28 (22.1)	0.3735
Cefepime	78 (21.9)	56 (24.1)	22(17.3)	0.1432
Ceftazidime	83 (23.3)	55 (23.7)	28 (22.1)	0.7939
Colistin	3 (0.8)	3 (1.3)	0 (0.0)	0.5550
Gentamicin	51 (14.3)	41 (17.7)	10 (7.9)	0.0112
Meropenem	82 (23.0)	57 (24.6)	25 (19.7)	0.3575
Piperacillin	85 (23.8)	61 (26.3)	24 (18.9)	0.1213
Ticarcillin	123 (34.5)	85 (36.6)	38 (29.9)	0.2448
Tobramycin	69 (19.3)	53 (22.8)	16 (12.6)	0.0244
Ciprofloxacin	73 (23.0)	58 (25.0)	15 (17.1)	0.1387
Pefloxacin	74 (25.7)	64 (27.6)	10 (17.5)	0.1306
MDR	70 (19.6)	53 (22.8)	17 (13.6)	0.0369
XDR	44 (12.3)	37 (15.9)	7 (5.6)	0.0040
DTR	41 (11.5)	36 (15.5)	5 (4)	0.0008

COVID-19: coronavirus-disease 2019; DTR: difficult-to-treat; MDR: multi-drug-resistant; XDR: extensively-drug-resistant.

**Table 5 antibiotics-12-00670-t005:** Antimicrobial resistance of *Pseudomonas aeruginosa* strains isolated from patients’ blood cultures compared to patients’ outcome at 30 days after the occurrence of bacteremia.

Antibacterial	Survived (%)	Died (%)	*p*
Amikacin	27 (11.1)	34 (29.8)	<0.0001
Aztreonam	42 (17.4)	48 (42.1)	<0.0001
Cefepime	34 (14.0)	44 (38.6)	<0.0001
Ceftazidime	37 (15.2)	46 (40.4)	<0.0001
Colistin	1 (0.4)	2 (1.8)	0.2401
Gentamicin	23 (9.5)	28 (24.6)	0.0003
Meropenem	36 (14.8)	46 (40.4)	<0.0001
Piperacillin	39 (16.0)	46 (40.4)	<0.0001
Ticarcillin	64 (26.3)	59 (51.8)	<0.0001
Tobramycin	30 (12.3)	39 (34.2)	<0.0001
Ciprofloxacin	35 (16.2)	38 (37.3)	<0.0001
Pefloxacin	37 (18.6)	37 (41.6)	<0.0001
MDR	33 (13.6)	37 (32.5)	<0.0001
XDR	15 (6.2)	29 (25.4)	<0.0001
DTR	13 (5.3)	28 (24.6)	<0.0001

DTR: difficult-to-treat; MDR: multi-drug-resistant; XDR: extensively-drug-resistant.

**Table 6 antibiotics-12-00670-t006:** Results of the regression analysis regarding patient mortality at 30 days after the occurrence of *Pseudomonas* spp. bacteremia.

Characteristic	Univariate Analysis *p*	Multivariate Analysis *p*	OR (95% CI)
Age (per year)	<0.0001	<0.001	1.032 (1.019–1.045)
Acquired in ICU	0.0073	0.01	2.213 (1.378–3.556)
Hospital day when the positive culture was drawn (per day)	0.0231	0.005	1.01 (1.003–1.017)
*P. aeruginosa*	0.0118	0.075	1.985 (0.933–4.224)
MDR	<0.0001	0.631	0.79 (0.302–2.065)
XDR	<0.0001	0.186	3.019 (0.587–15.527)
DTR	<0.0001	0.278	2.415 (0.491–11.865)

CI: confidence intervals; DTR: difficult-to-treat; ICU: intensive care unit; MDR: multi-drug-resistant; OR: odds ratio; XDR: extensively-drug-resistant.

## Data Availability

The data presented in this study are available on request from the corresponding authors.

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
