# Peer review of "Pseudomonas Bacteremia in a Tertiary Hospital and Factors Associated with Mortality"

_antibiotics, 2023, doi:10.3390/antibiotics12040670_

Round 1

Reviewer 1 Report

Major comments:

 1.    Line 20 / line 122 - The author states that they isolated 419 different strains. This is not true and the statement should be corrected.

2.  Line 84 – Nosocomial infection criteria can be controversial and questionable. The definition of nosocomial infection is not related to the fact when blood cultures were taken but to the onset of a clinical problem. Hospital-acquired infections are nosocomially acquired infections that are typically not present or might be incubating at the time of admission. These infections are usually acquired after hospitalization and manifest 48 hours after admission to the hospital!

3.   A data on where the blood cultures were taken is not relevant for the prognosis of sepsis. The prognosis of sepsis depends on the severity of the disease, the number of affected organs, comorbidity, etc. The least that needs to be added are scoring systems APACHE and SOFA.

4.      A data in table 1 / table 6 - (Day of culture positivity) is completely unclear and incomprehensible, the author should explain it more precisely.

5.      The author should explain how they defined pre and post covid time.

6.      Line 231 - In the discussion the risk factors of Ps. aeruginosa BSI should be described more precisely. Previous antimicrobial treatment  or hospitalization are risk factors for multidrug resistant bacterial infections.

7.      Line 233-235 – These are non-specific parameters that are not characteristic only for gram-negative sepsis.

8.      Line 275-277 - The message that colistin is an appropriate antibiotic for the treatment of BSI infections with resistant Ps. aeruginosa strains is completely incorrect and by no means as monotherapy. We have new antibiotics that are more suitable – ceftolozan/tazobaktam, ceftazidim/avibactam, cefiderokol, imipenem/cilastatin/relebaktam, aztreonam/avibaktam.

9.      A lenght of hospitalization before drawing the positive blood culture cannot be associated with higher mortality by itself.  The reasons are not defined. Clinical data are missing.

Minor comments:

1.      Line 24 / line 127 - …the median day the blood culture was drawn was 15. A vague statement. They were probably patients with nosocomial sepsis? This should be clearly recorded.

Author Response

Major comments:

  1. Line 20 / line 122 - The author states that they isolated 419 different strains. This is not true and the statement should be corrected.

Response: Thanks for the comment. We changed that as can be seen in the abstract and in the results section of the revised manuscript.

  1. Line 84 – Nosocomial infection criteria can be controversial and questionable. The definition of nosocomial infection is not related to the fact when blood cultures were taken but to the onset of a clinical problem. Hospital-acquired infections are nosocomially acquired infections that are typically not present or might be incubating at the time of admission. These infections are usually acquired after hospitalization and manifest 48 hours after admission to the hospital!

Response: Thanks for the comment. We do agree that hospital-acquired infections develop after 48 hours from admission, as community-acquired infections could manifest within the first 48 hours after admission. Since this is a manuscript based on data from microbiology, with an inability to draw clinical data, the only way to discriminate between hospital-acquired and community-acquired infections was to use the criterion of 48 hours. Thus, as explained in the methods section of the manuscript, blood cultures that were drawn 48 hours after admission (not blood cultures that turned positive after 48 hours – these could have been drawn earlier), were considered to be associated with a hospital-acquired infection. As blood cultures are taken as a result of clinical deterioration, most likely due to fever, we feel that this discrimination is at least adequately accurate. We have added a sentence in the limitations section, however, mentioning this.

  1. A data on where the blood cultures were taken is not relevant for the prognosis of sepsis. The prognosis of sepsis depends on the severity of the disease, the number of affected organs, comorbidity, etc. The least that needs to be added are scoring systems APACHE and SOFA.

Response: This is totally correct. Indeed, scoring systems based on clinical and laboratory characteristics of the patients have established value for prognostication. Unfortunately, this is a microbiologic study, and the few non-microbiological data shown are all that can be drawn from the electronic system. An attempt to find the clinical data that would be required to present the APACHE or SOFA scores would require at least months of work, while, at best, even more than 50% of the patients would still have missing data. This is why, by design, we focused primarily on presenting data on microbiology. From that perspective, one of the few pieces of data we had available from the electronic medical system was the site where patients had been hospitalized. Thus, we added that information to the analysis. We do understand that this is an epiphenomenon of the underlying clinical condition of the patients. We have added that information in the limitations subsection of the discussion section and elsewhere as can be seen in the discussion section of the revised manuscript.

  1. A data in table 1 / table 6 - (Day of culture positivity) is completely unclear and incomprehensible, the author should explain it more precisely.

Response: Thanks for the comment. We changed that into ‘Hospital day when the positive culture was drawn’

  1. The author should explain how they defined pre and post covid time.

Response: Post-COVID-19 era was defined as the era from 2020 until the end of the study. This is reasonable since the COVID-19 pandemic had its origins in December 2019. We have added this definition to the methods section of the revised manuscript.

  1. Line 231 - In the discussion the risk factors of Ps. aeruginosa BSI should be described more precisely. Previous antimicrobial treatment or hospitalization are risk factors for multidrug resistant bacterial infections.

Response: Thanks for the comment. We have revised that part in the discussion section and now we state the majority of the factors that are associated with the development of bacteremia by Pseudomonas and also added the appropriate references. This can be seen in the revised version of the manuscript.

  1. Line 233-235 – These are non-specific parameters that are not characteristic only for gram-negative sepsis.

Response: Indeed. We have changed that part to highlight that these findings may also occur in sepsis by other organisms as well and they are not specific for Gram-negative sepsis. This can be seen in the discussion section of the revised version of the manuscript.

  1. Line 275-277 - The message that colistin is an appropriate antibiotic for the treatment of BSI infections with resistant Ps. aeruginosa strains is completely incorrect and by no means as monotherapy. We have new antibiotics that are more suitable – ceftolozan/tazobaktam, ceftazidim/avibactam, cefiderokol, imipenem/cilastatin/relebaktam, aztreonam/avibaktam.

Response: Thanks for the comment. We have changed the discussion section to allow the reader to understand that these beta-lactam antibiotics, which had not been tested in the present study, as already mentioned in the limitations subsection of the discussion section, should be the first choice in susceptible Pseudomonas isolates, instead of colistin. This can be seen in the discussion section of the revised manuscript.

  1. A lenght of hospitalization before drawing the positive blood culture cannot be associated with higher mortality by itself. The reasons are not defined. Clinical data are missing.

Response: Thanks for the comment. Indeed, it may not be, and this could also be an epiphenomenon of the severity of the clinical condition of the patients, which, as stated above, was not the focus of the present study, and it is not possible to address it with the methodology we occupied. We have mentioned that in the discussion section as can be seen in the revised version of the manuscript.

Minor comments:

  1. Line 24 / line 127 - …the median day the blood culture was drawn was 15. A vague statement. They were probably patients with nosocomial sepsis? This should be clearly recorded.

Response: Thanks. We have changed those sentences to allow the reader to understand that the range is 0 to 267 days, thus, many bacteremia episodes are not hospital-acquired, just as mentioned at the beginning of those sentences. These changes can be seen in the revised version of the manuscript in the abstract and the results section.

Reviewer 2 Report

This manuscript is original. After the intensive use of antimicrobials in COVID_19 period, antimicrobial resitance of pathogens is a matter of curiosity. The article is interesting in this respect. It is useful to ısolation rate of Pseudomonas species for clinicians.

 The data of the study are detailed, statistical analysis is appropriate. The results are nicely explored and discussed in detail.

Here are few suggestions for work: It is recommended to stated that bacteremia cases are primary or secondery bacteremia in the manuscript.

Author Response

This manuscript is original. After the intensive use of antimicrobials in COVID_19 period, antimicrobial resitance of pathogens is a matter of curiosity. The article is interesting in this respect. It is useful to ısolation rate of Pseudomonas species for clinicians.

 The data of the study are detailed, statistical analysis is appropriate. The results are nicely explored and discussed in detail.

Response: Thanks for the nice comments.

Here are few suggestions for work: It is recommended to stated that bacteremia cases are primary or secondery bacteremia in the manuscript.

Response: Thanks for the comment. Since this study was focused on microbiological data, very few data regarding patients were collected (mainly ward where the blood culture was collected, hospital day where that occurred, gender, age, and outcome). Thus, since information regarding clinical characteristics, treatment, and whether bacteremia was primary or secondary are not available based on the methodology we used, we cannot add that in the manuscript. We have mentioned that in the limitations subsection of the discussion section of the revised manuscript.

Reviewer 3 Report

In this manuscript authors have addressed the prevalence and factors associated with mortality due to blood stream infections caused by various strains of Pseudomonas and primarily by P. aeruginosa

Line 140, 149 : Italicize species name

Line 145: "relative" is redundant

Line 169-174: More explanation is desirable as to why p value was so high/ less significant while discussing susceptibility to antibiotics pre/post COVID-19 era

Author Response

In this manuscript authors have addressed the prevalence and factors associated with mortality due to blood stream infections caused by various strains of Pseudomonas and primarily by P. aeruginosa

Line 140, 149 : Italicize species name

Response: Thanks for the comment. We corrected that in several parts of the manuscript as can be seen in the revised version.

Line 145: "relative" is redundant

Response: We corrected that.

Line 169-174: More explanation is desirable as to why p value was so high/ less significant while discussing susceptibility to antibiotics pre/post COVID-19 era

Response: Thanks for the comment. It is interesting that in many studies there is a relatively stable rate of resistant Pseudomonas strains even though an increasing rate of infections was noted. There were some theories regarding infection prevention and control measures that were more tightly applied during the COVID-19 pandemic, or a theory regarding a higher consumption of third-generation cephalosporins could drive an increase in cephalosporin-resistant but carbapenem-susceptible P. aeruginosa. However, there are no clear explanations, only assumptions. Thus, we have changed that part in the discussion section by mentioning these, and added some references. This can be seen in the discussion section of the revised manuscript.

Round 2

Reviewer 1 Report

Corrections, explanations and additions to the content are appropriate and acceptable.

Author Response

Thanks for the comment. We made some other modifications according to editorial comments that we feel have further improved the manuscript.